# A Novel Methodology for Predicting the Production of Horizontal CSS Wells in Offshore Heavy Oil Reservoirs Using Particle Swarm Optimized Neural Network

**Lijun Zhang [1,2], Haojun Xie [1,2], Zehua Fan [3,4], Yuting Bai [1,2], Jinpeng Hu [3,4], Chengkai Wang [3,4] and Xiaofei Sun [3,4,*]**

1   State Key Laboratory of Offshore Oil Exploitation, Beijing 100028, China
2   CNOOC Research Institute Co., Ltd., Beijing 100028, China
3   School of Petroleum Engineering, China University of Petroleum (East China), Qingdao 266580, China
4   Key Laboratory of Unconventional Oil & Gas Development, China University of Petroleum (East China), Ministry of Education, Qingdao 266580, China
*   Correspondence: xfsun@upc.edu.cn; Tel.: +86-13021678278

**Abstract:** Cyclic steam stimulation (CSS) is one of the main offshore heavy oil recovery methods used. Predicting the production of horizontal CSS wells is significant for developing offshore heavy oil reservoirs. Currently, the existing reservoir numerical simulation and analytical models are the two major methods to predict the production of horizontal CSS wells. The reservoir numerical simulation method is tedious and time-consuming, while the analytical models need many assumptions, decreasing models' accuracy. Therefore, in this study, a novel methodology combining the particle swarm optimization algorithm (PA) and long short-term memory (LM) model was developed to predict the production of horizontal CSS wells. First, a simulation model was established to calculate the cumulative oil production (COP) of horizontal CSS wells under different well, geological, and operational parameters, and then the correlations between the calculated COP and parameters were analyzed by Pearson correlation coefficient to select the input variables and to generate the initial data set. Then, a PA-LM model for the COP of horizontal CSS wells was developed by utilizing the PA to determine the optimal hyperparameters of the LM model. Finally, the accuracy of the PA-LM model was validated by the initial data set and actual production data. The results showed that, compared with the LM model, the mean absolute percentage error (MAPE) of the testing set for the PA-LM model decreased by 4.27%, and the percentage of the paired points in zone A increased by 2.8% in the Clarke error grids. In addition, the MAPEs of the training set for the PA-LM and LM models stabilized at 267 and 304 epochs, respectively. Therefore, the proposed PA-LM model had a higher accuracy, a stronger generalization ability, and a faster convergence rate. The MAPEs of the actual and predicted COP of the wells B1H and B5H by the optimized PA-LM model were 8.66% and 5.93%, respectively, satisfying the requirements in field applications.

**Keywords:** offshore heavy oil reservoirs; cyclic steam stimulation; particle swarm optimization algorithm; long short-term memory model; cumulative oil production

## 1. Introduction

The current pandemic, downturn, and low oil prices result in a larger global energy demand for hydrocarbon [1]. As the conventional onshore oil resources decrease, the interest in developing heavy oil resources from offshore reservoirs increases [2,3]. The proven heavy oil resources in Bohai Bay, China, are more than 4 billion tons [4]. Cyclic steam stimulation (CSS) is one of the main methods used for offshore heavy oil reservoirs [5].

Predicting the production of horizontal CSS wells is significant for developing offshore heavy oil reservoirs. Currently, the existing reservoir numerical simulation and analytical models are the two major methods to predict the production of horizontal CSS wells [6].

However, the reservoir numerical simulation is tedious and time-consuming, because it requires comprehensive data collection, a reliable simulation model, and history matching [7,8], while the analytical models need many assumptions to simplify the production process of horizontal CSS wells, decreasing the models' accuracy [9,10].

Recently, artificial intelligence algorithms, such as back propagation (BP) neural networks, support vector machines (SVM), and autoregressive integrated moving average (ARIMA) have been commonly used in the petroleum industry to solve some unsettled problems by conventional methods.

Yang et al. built an ARIMA model to predict oil well production [11]. Yuan et al. studied the profile control of the 32 wells of the Dongxin block by using the SVM method instead of numerical simulations [12] Based on the historical data of 15 oilfields in China, Hu et al. improved the BP neural network by the combination of fuzzy clustering algorithm and a genetic algorithm and used the network for oil production prediction [13]. Lu. et al. coupled the particle swarm optimization algorithm (PA) and machine learning to form a computational framework, which was used to predict shale oil production and optimize the fracturing parameters [14].

Compared with the aforementioned methods, recurrent neural networks (RNN) can predict time-series data by introducing the concept of timing sequence [15]. To solve the shortcomings of the traditional RNN model, such as gradient explosion and disappearance, Hochreater et al. developed a long short-term memory (LM) model by designing the cell state and gate structure in the RNN model [16,17]. Currently, the LM model has been successfully used in many areas, such as climate prediction, machine life prediction, and disease prediction [18–21].

Malhotra et al. developed an LM-based encoder–decoder scheme to estimate the remaining useful life of a machine [18]. Li et al. used the LM model for predicting the 24 h PM2.5 and confirmed that the LM model performed well with a short training time and low errors [19]. Kırbas et al. simulated the COVID-19 cases by using the ARIMA, RNN, and LM models and found that the LM model was the most accurate [20]. Kratzert et al. observed that the LM model could exactly characterize the trend of time-sequence data and was utilized to predict the rainfall-runoff [21].

At present, the LM method has not been used to predict the cumulative oil production (COP) of horizontal CSS wells in offshore heavy oil reservoirs. In addition, as mentioned above, the LM model solved the shortcomings of the RNN model by introducing the cell state and gate structure. However, this improvement also increases some hyperparameters in the LM model. Currently, the empirical method was commonly used to manually determine these hyperparameters, which results in low prediction accuracy. Therefore, the PA was utilized to determine the optimal hyperparameters of the LM model in this study. Then, a novel methodology combining the PA and LM model was developed to predict the COP of horizontal CSS wells in offshore heavy oil reservoirs located at Bohai Bay, China.

## 2. Methodology

The workflow of the methodology combining the PA and LM model for predicting the COP of CSS horizontal wells is presented in Figure 1, which includes the following processes. (1) Generation of the initial data set: the COP values of horizontal CSS wells under different geological, well, and operational conditions were calculated by an established simulation model. Then, the correlations between the COP and model parameters were analyzed by Pearson correlation coefficient to determine the input variables. Finally, the values of input variables and COP were normalized to form the initial data set. (2) Establishment of the PA-LM model: the PA-LM model was developed by using the PA to determine the optimal hyperparameters of the LM model. (3) Evaluation and application of the PA-LM model: the initial data set including the training and testing sets were used to train and validate the accuracy of the developed PA-LM model. Then, the developed PA-LM model was used to predict the COP of two typical horizontal CSS wells.

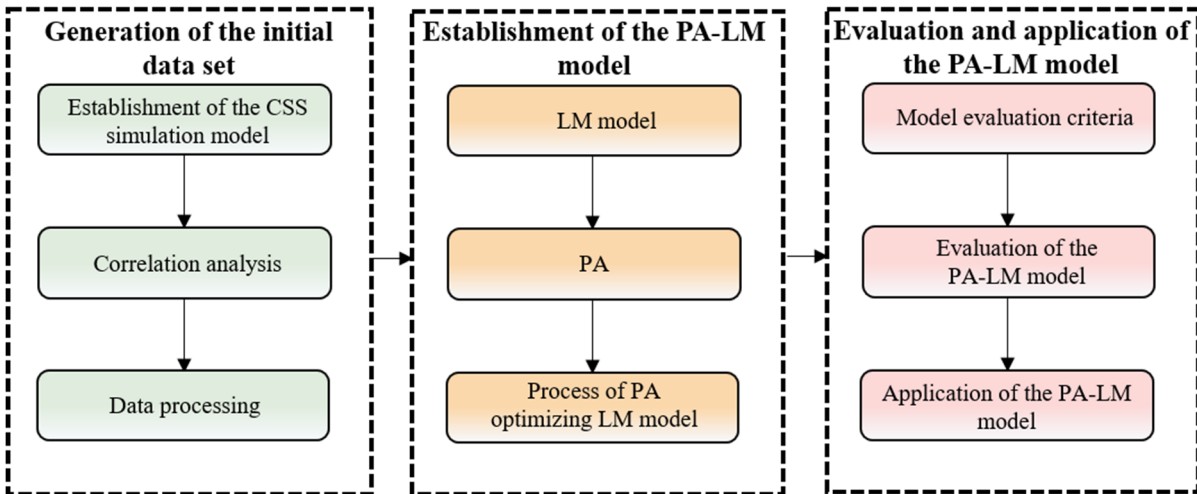

**Figure 1.** The workflow of the methodology combining the PA and LM model for predicting the COP of CSS horizontal wells.

## 3. Generation of the Initial Data Set

### 3.1. Establishment of the CSS Simulation Model

The STARS numerical simulator from CMG was used to build a typical CSS simulation model. The calculated COP of horizontal CSS wells under different geological, well, and operational conditions by the simulation model served as the initial data sets for the subsequent PA-LM model.

Figure 2 shows the established three-dimensional reservoir model and the horizontal well model. Table 1 shows the average values of the reservoir parameters, oil viscosities at different temperatures, oil and water relative permeability curves, heat transfer parameters of rocks and fluids, and operational parameters used in the CSS simulation model.

To ensure the accuracy of the established CSS simulation model and lay a solid foundation for generating the initial data sets for the subsequent PA-LM model, the COP values (primary production and the first cycle of the CSS process) of the horizontal CSS well were matched by modifying the model parameters, such as oil and water relative permeabilities, reservoir parameters, and heat transfer parameters of rocks and fluids within a reasonable range. The history matching results from Figure 3 indicate that the successful modification of the aforementioned parameters resulted in a satisfactory match for the horizontal CSS well.

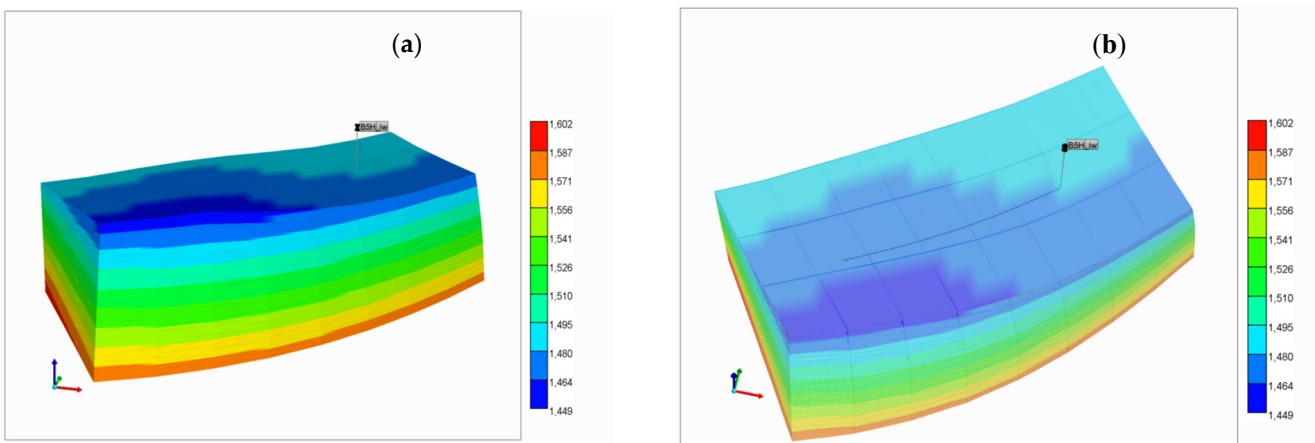

**Figure 2.** The established three-dimensional reservoir model and the horizontal well model: (**a**) three-dimensional reservoir model; (**b**) the horizontal well model.

**Table 1.** The basic parameters used in the CSS simulation model.

| Reservoir Parameters | Values |
| --- | --- |
| Reservoir porosity | 0.3 |
| Reservoir permeability (MD) | 3000 |
| Initial reservoir temperature (°C) | 54 |
| Oil saturation | 0.6 |
| **Well Parameters** | **Values** |
| Well spacing (m) | 200 |
| Well distance (m) | 200 |
| **Heat Transfer Parameters of Rocks and Fluids** | **Values** |
| Rock volumetric heat capacity (kJ/(cm$^3$·°C)) | $2.575 \times 10^6$ |
| Reservoir rock thermal conductivity (kJ/(cm·min·°C)) | $1.634 \times 10^5$ |
| Oil phase thermal conductivity (kJ/(cm·min·°C)) | $9.77 \times 10^3$ |
| Water phase thermal conductivity (kJ/(cm·min·°C)) | $5.99 \times 10^4$ |
| Gas phase thermal conductivity (kJ/(cm·min·°C)) | $1.9 \times 10^3$ |
| **Operational Parameters** | **Values** |
| Soaking time (day) | 5 |
| Steam injection volume (m$^3$) | 6600 |
| Steam injection rate (m$^3$/day) | 300 |
| Steam quality | 0.4 |
| liquid production rate (m$^3$/day) | 90 |
| Steam temperature (°C) | 340 |

| Oil Viscosities at Different Temperatures | |
| --- | --- |
| **Temperature (°C)** | **Viscosities (mPa·s)** |
| 20 | 74,704.94 |
| 50 | 3969.97 |
| 100 | 198.82 |
| 150 | 35.19 |
| 200 | 11.84 |
| 250 | 5.66 |
| 300 | 3.32 |

| Water and Oil Relative Permeabilities | | |
| --- | --- | --- |
| **$S_w$** | **$K_{rw}$** | **$K_{ro}$** |
| 0.305 | 0.000 | 1.000 |
| 0.373 | 0.006 | 0.738 |
| 0.427 | 0.012 | 0.572 |
| 0.497 | 0.036 | 0.380 |
| 0.505 | 0.045 | 0.360 |
| 0.533 | 0.060 | 0.292 |
| 0.588 | 0.080 | 0.173 |
| 0.626 | 0.100 | 0.104 |
| 0.720 | 0.147 | 0.000 |
| 1.000 | 1.000 | 0.000 |

After obtaining a suitable history match, 275 sets of simulations were performed to predict the COP of the horizontal CSS well under different well, geological, and operational conditions. The values of the well, geological, and operational parameters used in the simulations are within the parameter range of actual reservoirs in Bohai Bay, China. Figure 4 exhibits some of the calculated COPs of the horizontal CSS well at various conditions.

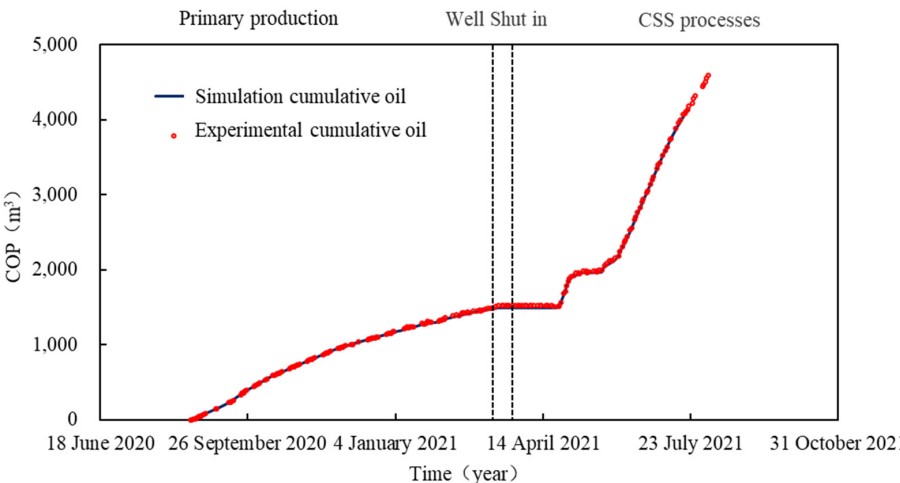

**Figure 3.** History matching of the COP values of the primary production and the first cycle of the CSS process of the horizontal CSS well.

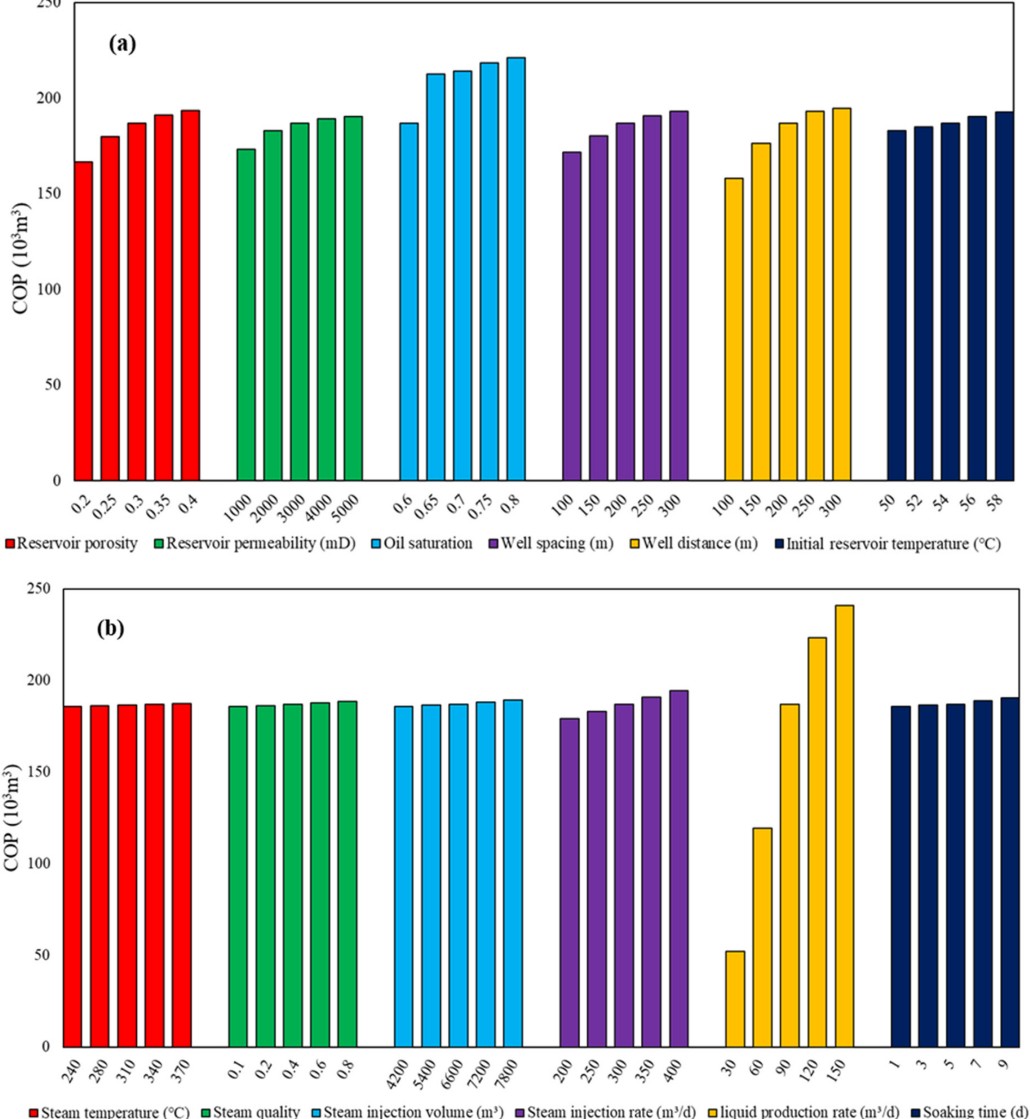

**Figure 4.** The calculated COP of the horizontal CSS well at various conditions: (**a**) well and geological conditions; (**b**) operational conditions.

### 3.2. Correlation Analysis

The quality and diversity of the calculated data have a decisive effect on the accuracy of the artificial intelligence model [22]. The Pearson correlation coefficient (*PCC*) was introduced to determine the relationships between the COP (*Y*) and geological and operational parameters (*X*). According to the paired data of COP and operational parameters {$(x_1, y_1)$, $\ldots$, $(x_n, y_n)$}, *PCC* can be calculated as follows [23]:

$$PCC = \frac{\sum_{i=1}^{n} (y_i - \bar{y})(x_i - \bar{x})}{\sqrt{\sum_{i=1}^{n} (y_i - \bar{y})^2 (x_i - \bar{x})^2}} \tag{1}$$

where *n* is the number of the paired data in the initial data set; $x_i$ and $y_i$ are the data points of operational parameters and *COP*, respectively, in the initial data set; $\bar{x} = \frac{1}{n} \sum_{i=1}^{n} x_i$, and analogously for $\bar{y}$. If the *PCC* is equal to 1, *X* and *Y* are linearly related. If the *PCC* is equal to 0, *X* and *Y* are not correlated.

It is clear from Table 2 that the liquid production rate, oil saturation, and reservoir permeability have strong correlations with the COP. The porosity, well spacing, well distance, steam temperature, and initial reservoir temperature have a medium relation with the COP. However, the *PCC* values between the soaking time, steam injection volume, steam injection rate, steam quality, and COP are less than 0.01, indicating that these parameters are weakly related to the COP. The correlation results are in line with the reality of horizontal CSS well production. Therefore, the selected input variables were the liquid production rate, reservoir permeability, oil saturation, porosity, well length, well spacing, steam temperature, and reservoir initial temperature.

**Table 2.** The calculated *PCC* values and the order of the parameters.

| Geological and Operational Parameters | PCC | Order |
|---|---|---|
| Reservoir porosity | 0.0171 | 4 |
| Reservoir permeability (MD) | 0.0217 | 3 |
| Oil saturation | 0.0557 | 2 |
| Well spacing (m) | 0.0160 | 5 |
| Well distance (m) | 0.0147 | 6 |
| Initial reservoir temperature (°C) | 0.0104 | 8 |
| Steam temperature (°C) | 0.0111 | 7 |
| Steam quality | 0.0022 | 11 |
| Steam injection volume (m$^3$) | 0.0045 | 10 |
| Steam injection rate (m$^3$/day) | 0.0051 | 9 |
| Liquid production rate (m$^3$/day) | 0.6321 | 1 |
| Soaking time (day) | 0.0018 | 12 |

### 3.3. Data Processing

To decrease the noise of the initial data set generated by the CSS simulation model, the normalization was conducted in the data processing processes, in which all of the values of input and output variables were transformed to the range of [0, 1] by the following Equation (2).

$$X' = \frac{X - X_{\min}}{X_{\max} - X_{\min}} \tag{2}$$

where the $X'$ and $X$ are the processed and initial values of the input and output variables, respectively. The $X_{\max}$ and $X_{\min}$ are the minimum and maximum data points. It is noted that the predicted COP by the developed PA-LM model needs to be rescaled to the original scale using the same parameters used for the normalization.

## 4. Establishment of the PA-LM Model

In this study, a novel model was developed by combing the PA and LM model to predict the COP of horizontal CSS wells. The aforementioned process is presented in Figure 5.

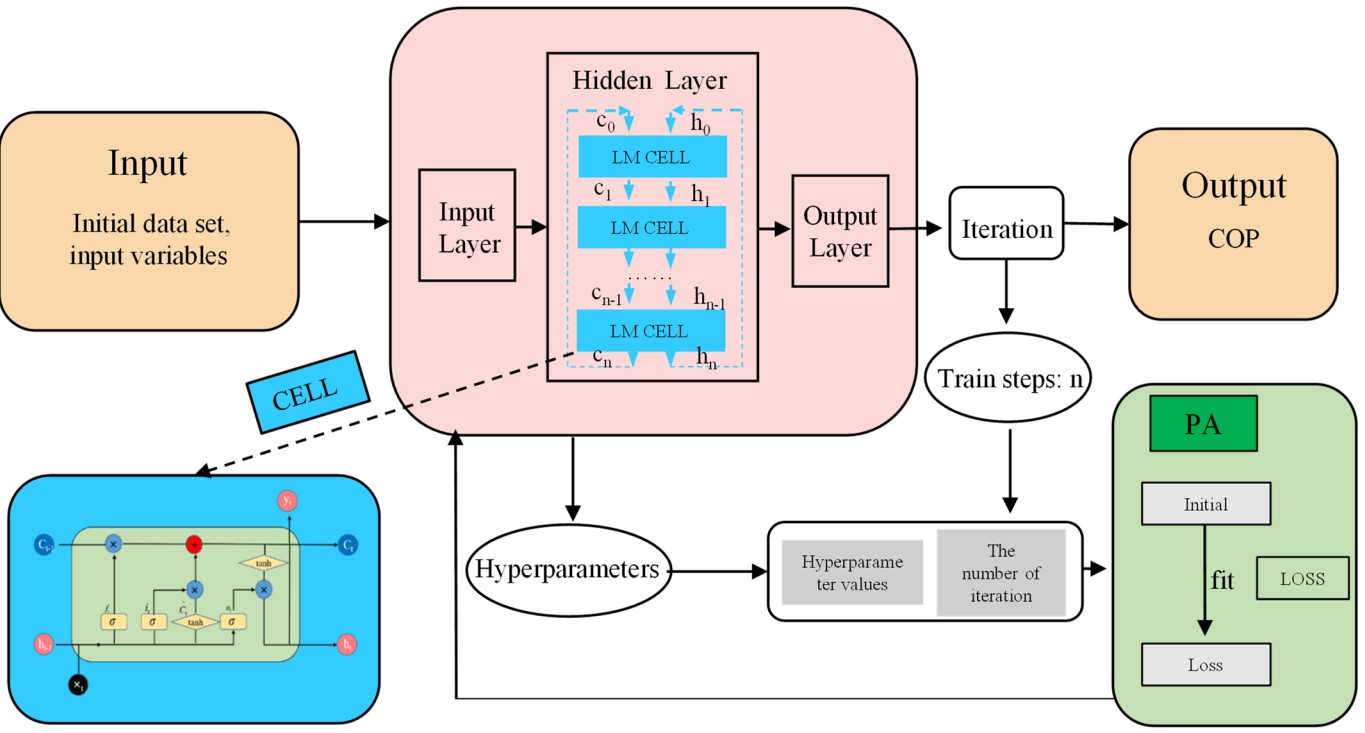

**Figure 5.** The establishment process of the PA-LM model.

### 4.1. LM Model

The architecture of LM model was developed through a trial phase in which we tested various model architecture settings such as layer count, size, and so on. The first layer is the input layer, including eight input variables, which are the well, geological, and operational parameters. The LSTM and dropout layers follow the first layer. A dense layer is used to encode the feature pattern of the input data and an output layer is the prediction of the COP. The addition of dropout layers improves the model performance by reducing overfitting [19].

The LM model was used to improve the RNN model by introducing the cell state and gate structures, as shown in Figure 6 [15,16]. When the initial data sets are input into the LM cells, the gate structures select appropriate information for the cell state. Therefore, the LM can be used to solve time-series problems.

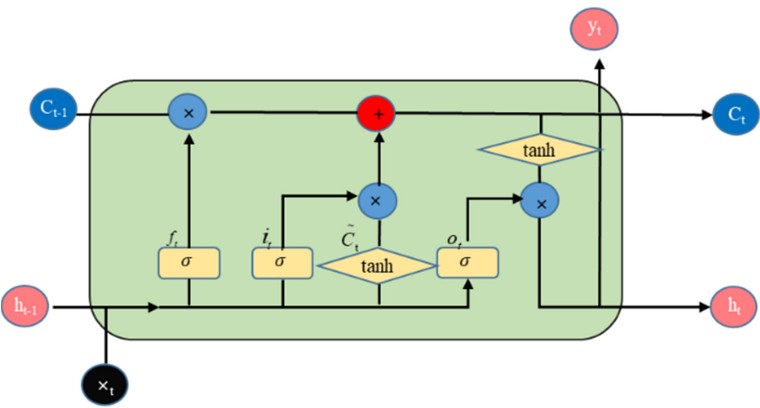

**Figure 6.** Schematic of a typical LM cell.

As shown in Figure 6, the symbols $\times$ and $+$ mean the multiplication and addition in the LM cell, respectively, and the arrow indicates the flow direction of initial data sets. The forget gate first applied to remove unnecessary information for the cell state is shown as follows:

$$f_t = \sigma([h_{t-1}, x_t] \times W_f + b_f) \tag{3}$$

where $\sigma$ is the activation function; $f_t$ is the forgetting threshold at time $t$; $W_f$ and $b_f$ are the weight and bias terms; $h_{t-1}$ means the output values at time $t-1$; and $x_t$ is the input values at time $t$.

The input gate can select and store information in the cell state from the current initial data sets. The specific expressions are shown below:

$$i_t = \sigma(_i[h_{t-1}, x_t] \times W + b_i) \tag{4}$$

$$\widetilde{C}_t = tanh([h_{t-1}, x_t] \times W_C + b_C) \tag{5}$$

where $i_t$ and $\widetilde{C}_t$ are the input threshold at time $t$; tanh is the activation function; $b_c$ and $b_i$ are bias terms; and $W_c$ and $W_i$ are the weights.

The cell state is updated as follows:

$$C_t =_t C_{t-1} \cdot \times f + \widetilde{C}_t \cdot \times i_t \tag{6}$$

where $C_{t-1}$ and $C_t$ are the cell state at time $t-1$ and $t$, respectively.

The output gate is used to output information at time $t$, which is written below:

$$o_t = \sigma([h_{t-1}, x_t] \times W_o + b_o) \tag{7}$$

where $o_t$ denotes the output threshold at time $t$; $W_o$ and $b_o$ are the weights and bias terms, respectively.

The output value $h_t$ and $y_t$ at time $t$ can be obtained as follows:

$$h_t = y_t = o_t \times tanh(C_t) \tag{8}$$

### 4.2. PA

In this study, the eight hyperparameters of the LM model, including the number of hidden layers ($n_1$), mini-batch size ($m$), number of neurons in hidden layers ($n_2$), learning rate ($L$), max epochs ($M$), learning rate drop factor ($F$), learning rate drop period ($P$), and dropout rate ($D$), were treated as the optimized parameters of PA in eight dimensions. The fitness value as the optimization objective was calculated to achieve the globally optimal values (the optimal hyperparameters). The position and velocity values of a particle can be calculated as follows.

$$V_{i,j}^{k+1} = wV_{i,j}^{k} + c_1 r\left(pbest_{i,j}^{k} - X_{i,j}^{k}\right) + c_2 r\left(gbest_{i,j}^{k} - X_{i,j}^{k}\right) \tag{9}$$

$$X_{i,j}^{k+1} = X_{i,j}^{k} + V_{i,j}^{k} \tag{10}$$

where $r$ is a random number; $c_1$ and $c_2$ are the learning factors; $w$ is the weight; $V_{i,j}^{k}$, $X_{i,j}^{k}$, $pbest_{i,j}^{k}$, and $gbest_{i,j}^{k}$ are the velocity values, position values, individual optimal values, and global optimal values of the $i$th particle in the $j$th dimension and the $k$th iteration, respectively, and $X_{i,j}^{k+1}$ and $V_{i,j}^{k+1}$ are the position and velocity values of the $i$th particle in the $j$th dimension and the $k+1$th iteration.

In this study, a linear method was used to calculate the inertia weight, which can effectively enhance the optimization ability of the PA.

$$\omega_t = \omega_{\max} - \frac{p(\omega_{\max} - \omega_{\min})}{T_{\max}} \tag{11}$$

where $\omega_{\min}$ and $\omega_{\max}$ are the minimum and maximum inertia weights, respectively; $T_{\max}$ is the maximum iteration number; $p$ is the current generation number; and $\omega_t$ is the $t$th inertia weights.

### 4.3. Process of PA Optimizing LM Model

Firstly, the aforementioned eight hyperparameters of the LM model were selected as the optimized parameters of the PA, and the position values of all the particles were randomly initialized within the ranges of the eight hyperparameters. Then, the LM model was developed and then trained by the training data set. The detailed processes are shown below.

(1) Initialize the particle parameters, including the number of particles $n$, $c_1$, $c_2$, $r$, and $T_{\max}$ in the PA. (2) A particle $X_i,O$ ($n_1$, $n_2$, $m$, $L$, $M$, $P$, $F$, and $D$) was randomly generated. The values of the particle $X_i,O$ were assigned to the hyperparameter of the LM model. (3) The initial data set was used to train and test the LM model and the corresponding COP values were calculated. The fitness value $fit_i$ of the $X_i,O$ was calculated as follows:

$$fit_i = 0.5^* \sum_{j=1}^{J} \frac{\left|\hat{y}^j - y^j\right|}{y^j} * \frac{1}{J} + 0.5^* \sum_{k=1}^{K} \frac{\left|\hat{y}^k - y^k\right|}{y^k} * \frac{1}{K} \tag{12}$$

In Equation (12): $y^k$ and $y^j$ are the COP values in the testing and training sets, respectively, $\hat{y}^k$ and $\hat{y}^j$ are the predicted COP values by the PA-LM model in the testing and training sets, and $J$ and $K$ are the size of the training and testing sets, respectively.

(4) The $pbest_{i,j}^k$, was initialized as the particle position in the initial state, and the $gbest_{i,j}^k$ was taken as the particle position with the lowest $fit_i$. (5) During each iteration, the velocity and position values of the particles were updated based on the $pbest_{i,j}^k$, and $gbest_{i,j}^k$ was calculated by Equations (9) and (10). Then, the new $fit_i$ of the particles were calculated and the $pbest_{i,j}^k$ and $gbest_{i,j}^k$ of the particles were updated. (6) After $T_{\max}$ was reached, the $gbest_{i,j}^k$ (the corresponding hyperparameters) were the optimal values for the LM model.

## 5. Evaluation and Application of the PA-LM Model
### 5.1. Model Evaluation Criteria

To verify the reliability of the model for predicting the COP of horizontal CSS wells under different conditions, the root mean squared error (RMSE), mean absolute percentage error (MAPE), mean squared error (MSE), and $R^2$ were selected as the model evaluation indicators, which are expressed as follows:

$$MAPE = \frac{1}{n}\sum_{i=1}^{n}\left|\frac{y_i - \hat{y}_i}{\hat{y}_i}\right| \tag{13}$$

$$MSE = \frac{1}{n}\sum_{i=1}^{n}(\hat{y}_i - y_i)^2 \tag{14}$$

$$RMSE = \sqrt{\frac{1}{n}\sum_{i=1}^{n}(\hat{y}_i - y_i)^2} \tag{15}$$

$$R^2 = 1 - \frac{\sum_{i=1}^{n}(y_i - \hat{y}_i)^2}{\sum_{i=1}^{n}(\overline{y} - \hat{y}_i)^2} \tag{16}$$

where $y_i$ and $\hat{y}_i$ are the COP value predicted by the model and COP value in the initial data set; $i$ is the indicator of data points; and $n$ is the total number of data points in the initial data set. A lower value of RMSE, MSE, and MAPE and a higher value of $R^2$ indicate a higher accuracy of the model.

### 5.2. Evaluation of the PA-LM Model

According to the aforementioned results of PCC, the liquid production rate, reservoir permeability, oil saturation, porosity, well length, well spacing, steam temperature, and reservoir initial temperature were highly related to the COP of horizontal CSS wells. Therefore, the aforementioned eight parameters were selected as the input variables, and the corresponding COP was utilized as the output variable for the model.

Based on the initial values of the input variables and the corresponding output variable (COP), the processed values of the input variables and output variable were obtained by the data processing (Equation (2)) and used as the initial data set. To build the PA-LM model and verify its reliability, 80% and 20% of data were randomly selected from the initial data set as a training set and a testing set. The key parameters used in the PA-LM model are shown in Table 3.

**Table 3.** The key parameters used in the PA-LM model.

| Hyperparameters of the LM Model | Values |
|---|---|
| $n_1$ | 250 |
| Number of input variables | 8 |
| Number of output variables | 1 |
| $n_2$ | 200 |
| $m$ | 15 |
| $L$ | 0.01 |
| Loss function | MAPE |
| Optimizer | Adam |
| $M$ | 400 |
| $P$ | 20 |
| $F$ | 0.8 |
| $D$ | 0.3 |
| **PA Parameters** | **Values** |
| $n$ | 20 |
| $c_1$ and $c_2$ | 1.5 |
| $T_{\max}$ | 300 |
| $r$ | [0, 1] |

During the training process, the hyperparameters of the LM model were optimized by the PA until the $T_{\max}$ was reached. Figure 7a presents the loss decline curves of the LM and PA-LM models in the training processes. As shown in Figure 7a, the MPAE values of the PA-LM model decrease with the increase in the number of epochs, confirming that a reliable PA-LM model for the testing process was formed. A comparison of the loss decline curves shown in Figure 7a indicates that the MAPEs stabilize at 267 and 304 epochs and the final MAPEs are 2.34% and 7.68% for the PA-LM and LM models, respectively. Therefore, the PA-LM model has a faster convergence rate and a higher accuracy than the LM model because the PA effectively improved the prediction accuracy. Figure 7b shows that the loss decline curves of the LM model and PA-LM model in the testing processes have a similar trend with those in the training processes.

The Clarke error grid method was also used to further verify the reliability of the PA-LM model [15].

The results from Figure 8 show that, for the LM model, 97.2% of the paired points were in the acceptable zones A, 2.1% in B, and 0.7% in zone C and D. No paired point ended up in zones E or F. For the PA-LM model, 100% of the paired points were in zone A, which was significantly more compared with the LM model. There were no paired points in zone B, C, D, or E.

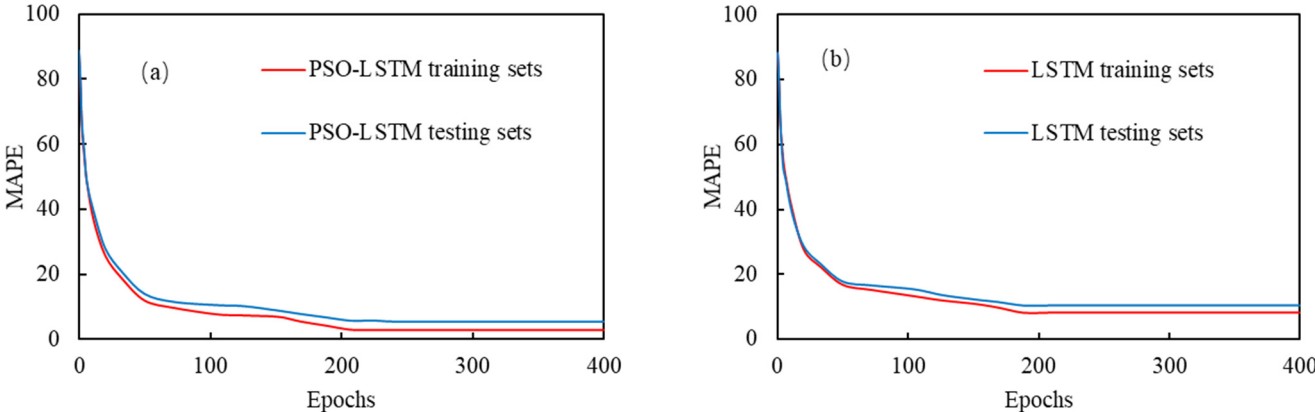

**Figure 7.** Comparisons of loss decline curve of LM and PA-LM model: (**a**) the training processes; (**b**) the testing processes.

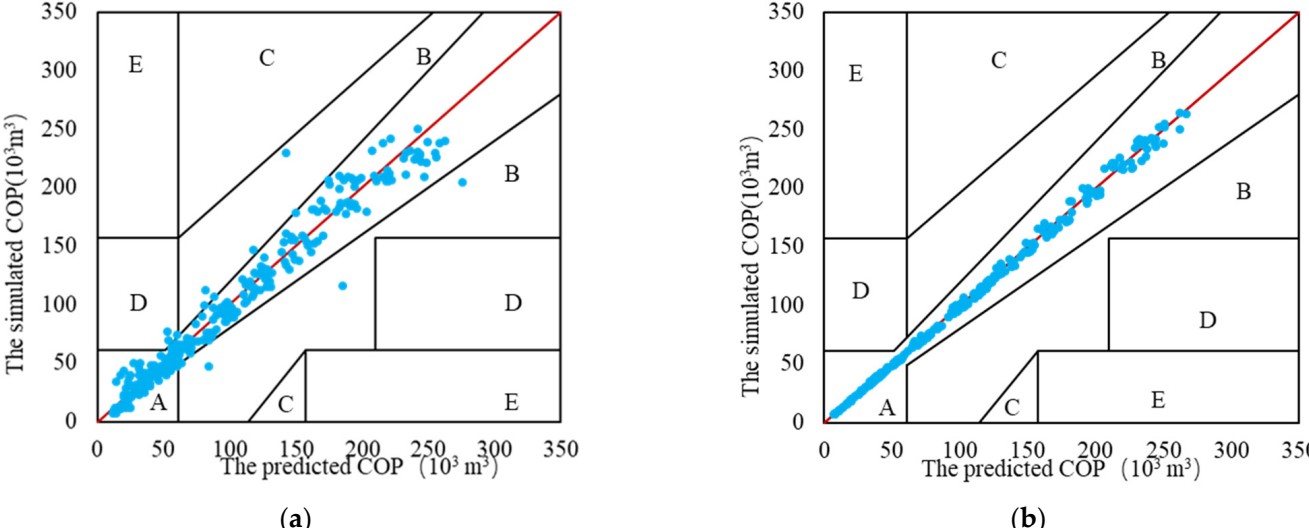

**Figure 8.** The Clarke error grids of the LM and PA-LM model: (**a**) LM model; (**b**) PA-LM model.

The paired points in zone A mean that the COP values predicted by the models do not deviate from the COP values determined by simulations by more than 20%. A comparison of the results in Figure 8 shows that the PA-LM model achieved a higher accuracy due to the higher percentage of paired points in zone A.

In addition, four typical data were randomly selected from the initial data set and compared with those predicted by the LM and PA-LM models (Figure 9). The results show that the COP values predicted by the PA-LM model fit well. However, the COP values predicted by the LM model deviated greatly from those in the initial data set. Therefore, the PA-LM model can predict the COP of horizontal CSS wells more accurately.

The four indicators of the training and testing sets for the two models, MAPE, MSE, RMSE, and $R^2$, are listed in Table 4. The lower values of MAPE, MSE, and RMSE as well as the higher values of $R^2$ of the training and testing sets indicate that the PA-LM model has higher accuracy than the traditional LM model.

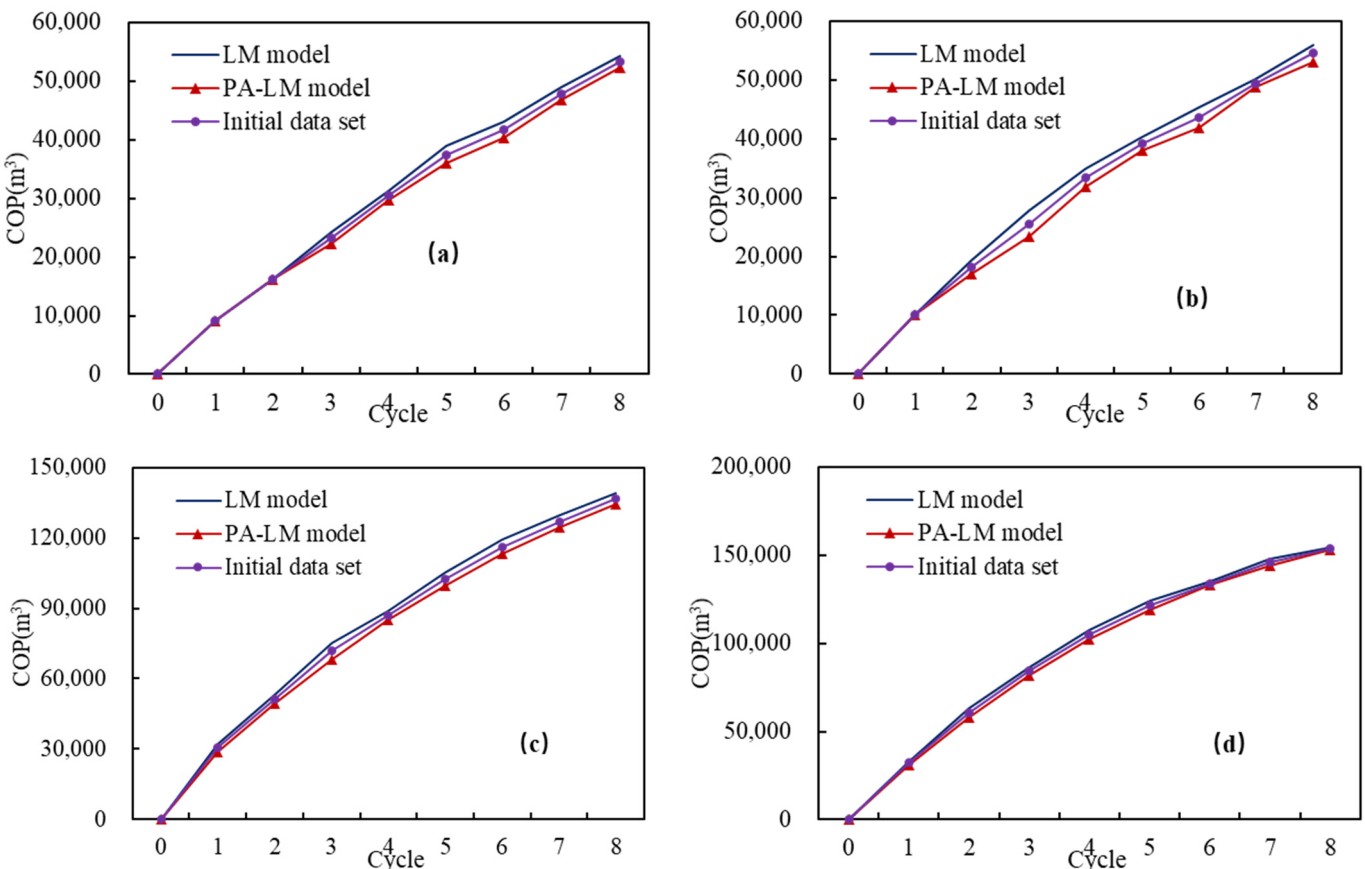

**Figure 9.** Comparisons of the COP values predicted by the LM and PA-LM models: (**a**) typical data 1; (**b**) typical data 2; (**c**) typical data 3; (**d**) typical data 4.

**Table 4.** Comparisons of MAPE, MSE, RMSE, and $R^2$ of the initial data sets for the LM and PA-LM models.

| The Types of Data Sets and Model | MAPE (%) | MSE | RMSE | $R^2$ |
|---|---|---|---|---|
| The training set of the LM model | 7.68 | $8876.4^2$ | 8876.4 | 0.9135 |
| The testing set of the LM model | 9.33 | $10,124.8^2$ | 10,124.8 | 0.9027 |
| The training set of the PA-LM model | 2.34 | $3931.3^2$ | 3931.3 | 0.9913 |
| The testing set of the PA-LM model | 5.06 | $6054.33^2$ | 6054.33 | 0.9807 |

### 5.3. Application of the PA-LM Model

In this section, we showed two field applications of the established PA-LM model. Two typical horizontal wells of CSS (B1H and B5H) were selected from the Lvda 21-2 reservoirs in Bohai Bay, China. These two horizontal wells only conducted one cycle of CSS. Therefore, the COP values of the first cycle of the horizontal wells were utilized to evaluate the PA-LM model's accuracy. As shown in Figure 10, good agreements between the predicted and actual COP values are obtained in the two cases. Table 5 shows that the MAPEs of the COP values for the two wells are 8.66% and 5.93%, respectively. Although the geological, well, and operational parameters are different for the two horizontal wells, the optimized PA-LM model presents the reliable predictions in the two horizontal wells. The results indicate that the developed PA-LM model has a good generalization. In conclusion, the aforementioned evaluations based on the initial data set and actual production data proved the PA-LM model's accuracy.

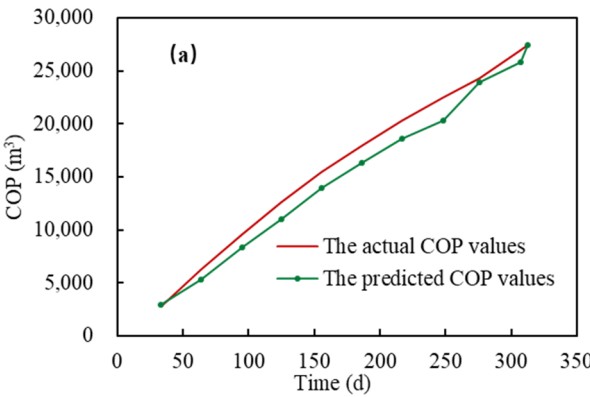 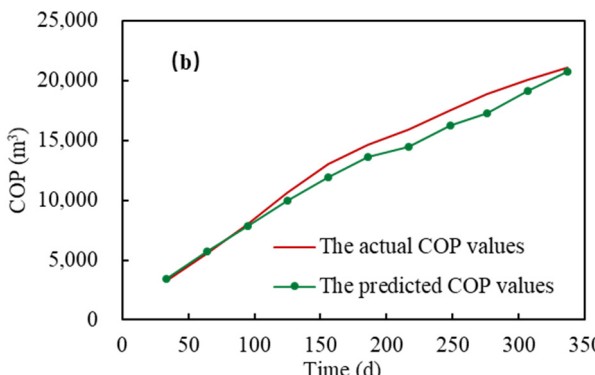

**Figure 10.** Comparison of the actual and predicted COP of the first cycle of the two horizontal wells: (**a**) well B1H; (**b**) well B5H.

**Table 5.** Evaluation of the accuracy of the PA-LM model by the actual COP values of the two horizontal CSS wells.

| Well Name | MAPE (%) | MSE | RMSE |
|-----------|----------|-----|------|
| B1H | 8.66 | $1363.81^2$ | 1363.81 |
| B5H | 5.93 | $1058.66^2$ | 1058.66 |

However, it is noted that more COP data need to be collected from the following cycles or other CSS wells to further verify the PA-LM model's accuracy with the subsequent development of the reservoirs in Bohai Bay, China. In addition, the developed PA-LM model in this study can only be used to predict the COP values of horizontal CSS wells in Bohai Bay, China, because the model was developed based on the parameters of actual reservoirs in Bohai, Bay, China. When the method is used in other offshore heavy oil reservoirs, the establishment process of the PA-LM model shown in this study should be re-conducted based on the practical reservoir conditions.

Figure 11 shows the predicted COP values of the two CSS horizontal wells in the eight cycles by the optimized PA-LM model. The predicted COP values of the eight cycles of two CSS horizontal wells are 226,753 and 147,117 $m^3$, respectively, which are expected to provide important references for decision-makers in developing offshore heavy oil reservoirs.

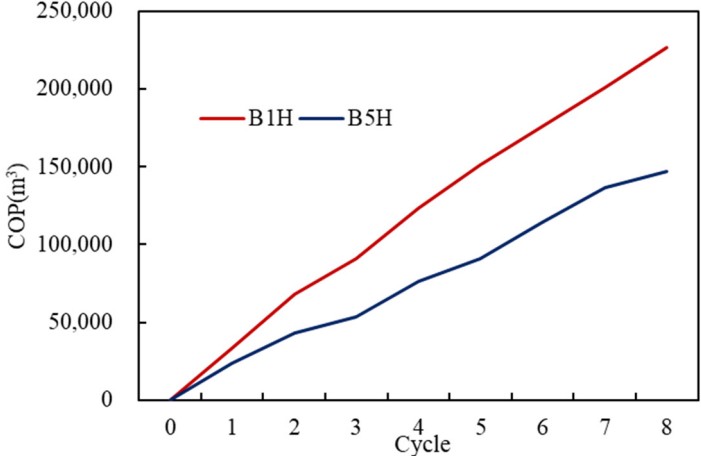

**Figure 11.** Prediction of the COP values of the two CSS horizontal wells by the optimized PA-LM model.

## 6. Conclusions

(1) Compared with the LM model, the MAPE of the testing set for the PA-LM model decreased by 4.27%, and the percentage of the paired points in zone A increased by

2.8% in the Clarke error grids. In addition, the MAPEs of the training sets for the PA-LM and LM models stabilized at 267 and 304 epochs, respectively. Therefore, the developed PA-LM model had a higher accuracy, a stronger generalization ability, and a faster convergence rate.

(2) The PA can be used to determine the optimal hyperparameters of the LM model. Therefore, the PA-LM model outperform the LM model in predicting the COP of horizontal CSS wells in offshore heavy oil reservoirs.

(3) The MAPEs of the predicted and actual COP values of the first cycle of the wells B1H and B5H (two horizontal CSS wells in Bohai Bay, China) by the optimized PA-LM model are 8.66% and 5.93%, respectively, satisfying the requirements in field applications. More COP data need to be collected from the following cycles or other CSS wells to further verify the PA-LM model's accuracy with the subsequent development of the reservoirs in Bohai Bay, China.

(4) The developed PA-LM model provides an effective method for predicting the COP values of horizontal CSS wells in Bohai Bay, China, which was validated by the initial data set and actual production data. However, when the method is used in other offshore heavy oil reservoirs, the establishment process of the PA-LM model shown in this study should be re-conducted based on the practical reservoir conditions.

(5) According to the calculated PCC results, the liquid production rate, oil saturation, and reservoir permeability are strongly related to the COP of the horizontal CSS wells. The soaking time, steam injection volume, steam injection rate, and steam quality have weak correlations with the COP of the horizontal CSS wells.

**Author Contributions:** Conceptualization, X.S. and L.Z.; methodology, Z.F. and H.X.; formal analysis, J.H. and C.W.; investigation, Y.B. and H.X.; writing—original draft preparation, X.S. and Z.F.; writing—review and editing, L.Z. All authors have read and agreed to the published version of the manuscript.

**Funding:** This research was funded by a project from Beijing Research Center of CNOOC (China) Co., LTD. (CCL2021RCPS0297RSN) and the Major Science and Technology of China National Offshore Oil Corporation (YXKY-ZX 06 2021).

**Institutional Review Board Statement:** Not applicable.

**Informed Consent Statement:** Not applicable.

**Data Availability Statement:** Data sharing is not applicable to this article.

**Conflicts of Interest:** The authors declare no conflict of interest.

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
