# Peer review of "A Novel Methodology for Predicting the Production of Horizontal CSS Wells in Offshore Heavy Oil Reservoirs Using Particle Swarm Optimized Neural Network"

_applsci, doi:10.3390/app13042540_

Round 1

Reviewer 1 Report

I commend the authors for their efforts in putting this manuscript together. No doubts, applications of an optimised neural network for production forecasting, especially for heavy oil reservoirs, is still in its infancy and studies such as this one, will add to the knowledge base in this area. Overall, this manuscript is technically sound, but I don't think it merits being published in its current form. I would recommend it for publication subject to the authors' corrections/answers to the following comments:

1. The title needs to be modified to reflect the full meaning of PA-LM e.g. "A novel methodology for predicting the production of horizontal CSS wells in offshore heavy oil reservoirs using Particle-Swarm Optimised Neural Network"

2. The authors should mention, in the abstract, at least one example of the "many challenges" facing the existing reservoir numerical simulation method and the analytical models. I know some were mentioned in the introduction, but it is also good to include at least one in the abstract. That will give an idea into the actual problem the authors planned to solve with the proposed model.

3. The RMSE and MSE values are obviously significant, about the same order as the COP values. It would be good to know how the authors arrived at the R2 values. The results shown in Figures 9 & 10 also show the model deviates significantly from the actual production profile.  Can the authors justify how this model is a good alternative to the numerical simulation, which as shown in Figure 3 gave a near perfect match with the actual production data.

4. Overall, I believe the authors need to revisit their model or performance metrics. I suggest re-running the model for a little longer (more epochs), apply drop-out between layers, apply data transformations, etc. I am certain there are tons of references materials the authors can leverage to do this.

5. The authors should change the LM-PSO in their first conclusion to PA-LM for consistency. I am not sure LM-PSO was mentioned anywhere else in the manuscript.

Reviewer 2 Report

Good quality article, the topic is relevant not only for a broad academic community, but also for practitioners and other relevant stakeholders.  The structure of the article is consistent with the requirements (the abstract disclose the contents of the main research results; the illustrations (tables, figures) are relevant). The article includes enough scientific analysis of literature and other sources on the subject of research. The research methods used are dequate and sufficient for this topic the results of the research reliable and well-founded. The conclusions in the article correct and logically grounded.

Reviewer 3 Report

The paper presented a methodology combing the particle swarm optimization algorithm and long and short-term memory model to predict the production of horizontal CSS wells. However, some problems still need to be addressed and explained in this current manuscript.

11. What are the major limitations of this model? Could you please highlight it in one section?

22.  Table 1 provides steam temperature equal to the reservoir temperature. Any reason for injection steam to be similar to reservoir temperature? Reservoir pressure data is not provided. What are the fluid injection phases, steam or hot water?

33. While COP is cumulative oil production, is water production been part of liquid production? What is the water cut during cyclic steam stimulation? Therefore, which parameter strongly correlates with COP, liquid production rate or oil production rate?

44. What are the data set to build PA-LM model? It is mentioned as a training and testing set randomly selected from the initial data set.

55. Why is the number of cycles not included in the input variables?

Round 2

Reviewer 1 Report

Although my comments about the error margins are yet to be addressed in the revised manuscript, I am happy to accept in the current form considering the other aspects of the manuscript are sound.